# VICINAL ASSESSMENT OF MODEL GENERALIZATION

## ABSTRACT

This paper studies how to assess the generalization ability of classification models on out-of-distribution test sets without relying on test ground truths. Existing works usually compute an unsupervised indicator of a certain model property, such as confidence and invariance, which is correlated with out-of-distribution accuracy. However, these indicators are generally computed based on a *single* test sample in isolation (and subsequently averaged over the test set), and thus are subject to spurious model responses, such as excessively high or low confidence. To address this issue, we propose to integrate model responses of *neighboring* test samples into the correctness indicator of the every test sample. Intuitively, if a model consistently demonstrates high correctness scores for nearby samples, it becomes more likely that the target sample will also be correctly predicted, and vice versa. This score is finally averaged across all test samples to indicate model accuracy holistically. This strategy is developed under the vicinal risk formulation, and, since its computation does not rely on labels, is called vicinal risk proxy (VRP). We show that VRP methodologically can be applied to existing generalization indicators such as average confidence and effective invariance and experimentally brings consistent improvements to these baselines. That is, stronger correlation with model accuracy is observed, especially on severe out-of-distribution test sets.

## 1 INTRODUCTION

Because of the ubiquitous existence of distributional shifts in real-world systems, it is important to evaluate the generalization capacity of trained models on out-of-distribution (OOD) test data. In practical OOD scenarios, because obtaining test ground truths is expensive, model evaluation techniques that do not rely on test labeled are attracting increasing attention.

For this problem, unsupervised risk measurements are introduced that capture useful model properties, such as confidence and invariance. These measurements serve as indicators of a model's generalization ability. Importantly, for a sample of interest without ground-truth, these methods compute a risk proxy *merely using this sample alone*. For example, Hendrycks and Gimpel (2017) use the maximum Softmax value of a test sample itself as its confidence score, and show that once averaged over the OOD test set, it serves as a reliable indicator. The Effective Invariance score (Deng et al., 2022) is computed as the prediction consistency between this sample and its transformed version (*e.g.*, rotation and grey-scale). Because these indicators measure model properties that underpin its generalization ability, they generally exhibit fair correlation with the model's out-of-distribution accuracy.

However, we find this isolated way of measuring model effectiveness for a sample of interest suffers from spurious model responses. In Fig. 1, we show two examples where an incorrect prediction on a test sample may have a much higher confidence score than a correct prediction. In other words, a high (*resp.* low) confidence or invariance score sometimes does not mean a correct (*resp.* incorrect) prediction. These erroneous scores, once accumulated in the test set, would compromise the effectiveness of risk measurements.

To address this problem, when computing the risk proxy for a test sample, we propose to integrate into its risk proxy the model behaviour on its adjacent samples, where such integration is performed using the vicinal distribution for the test sample (Fig. 1). Intuitively, if neighboring samples generally exhibit high risks (*e.g.*, low confidence), the center sample with excessively low risk will be assigned an increased risk score, and vice versa. Here, the contribution of each neighboring sample to the center sample is proportional to their similarity. This strategy allows model responses (risk proxy

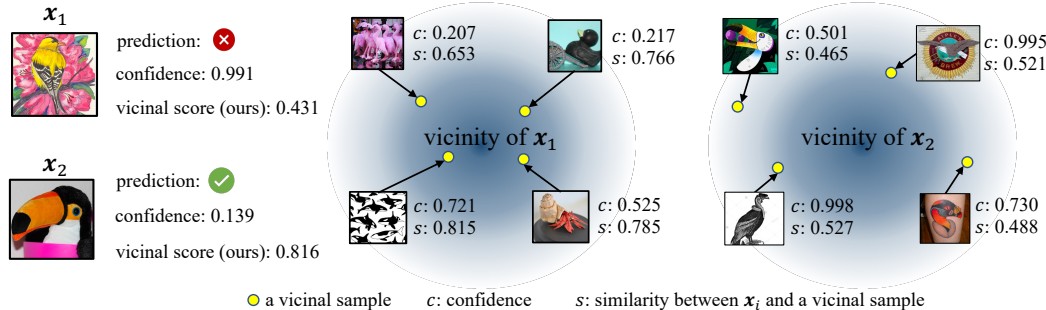

Figure 1: **Examples of spurious model responses and an intuitive illustration how our method corrects them.** Here we use confidence as model generalization indicator and aim to improve it. $x_1$ and $x_2$, from ImageNet-R (Hendrycks et al., 2021a), are classified correctly and incorrectly, respectively. However, their confidence scores are excessively high (0.991) and excessively low (0.139), respectively, indicating spuriousness. More reasonable scores (0.431 and 0.816) are given by the proposed vicinal method, which is a similarity weighted sum of confidence. For example, the vicinal score of $x_1$ is computed as: $0.431 = \frac{0.653 \times 0.207 + 0.766 \times 0.217 + 0.815 \times 0.721 + 0.785 \times 0.525}{0.653 + 0.766 + 0.815 + 0.785}$.

score) to better indicate prediction correctness for the test sample, as shown in Fig. 2 (b) and (d). To indicate the overall generalization ability of models, we further compute the vicinal risk proxy (VRP) as the average individual vicinal score over the entire test set.

Another advantage of this vicinal assessment scheme is that it can be applied on top of various risk proxies based on individual test samples, including confidence, invariance and their variants. Our experiments show that VRP brings consistent improvements to them: stronger correlation between risk proxies rectified by VRP and model OOD accuracy over 200 different classifiers is generally observed on 9 benchmarks. In summary, this paper has the following main points.

- We examine existing methods in OOD generalization assessment in the lens of risk estimation.

- We propose to integrate vicinal distribution of a sample into its risk estimate, to inhibit spurious model responses.

- The proposed vicinal risk proxy (VPR), when applied to existing risk proxies, brings consistent improvement: stronger correlation is observed between vicinal proxies and model OOD accuracy.

## 2 RELATED WORK

**Data-centric model generalization assessment** aims to predict the accuracy of a *given* model on *various* unlabeled test sets. Average model confidence (Hendrycks and Gimpel, 2017; Tu et al., 2023) on the testing samples is a simple and useful indicator of model accuracy. Guillory et al. (2021) propose using the confidence discrepancy between the validation and test sets to correct the confidence-based accuracy. Deng et al. (2021) tackle this challenge by comparing models based on their accuracy in self-supervised tasks. Garg et al. (2022) predict accuracy by using the percentage of testing samples exceeding a threshold learned from a validation set in the source domain. In addition to confidence, domain shift can also be used as a cue to predict model accuracy on the target set (Guillory et al., 2021; Deng and Zheng, 2021). This paper does not focus on this setup and only provides some results in the supplementary materials.

**Model-centric generalization assessment.** Some works focus on *in-distribution* generalization (Garg et al., 2021; Jiang et al., 2021; Negrea et al., 2020; Zhou et al., 2020). This paper instead studies *OOD* generalization. In this problem, we train a variety of models on a training set and predict and compare their performance on an unlabeled out-of-distribution test set. Deng et al. (2022) propose effective invariance (EI) to measure the consistency between model predictions on the original image and its transformed versions. It is also feasible to use data-centric indicators such as average confidence (Hendrycks and Gimpel, 2017; Tu et al., 2023). However, we find these methods

Figure 2: **Examples how vicinal risks of individual samples better separate models making a correct/incorrect prediction.** We use a single test sample from ImageNet-R and 140 models trained on the ImageNet training set. From (a) to (d), we use confidence (Tu et al., 2023; Hendrycks and Gimpel, 2017), confidence + our method, EI (Deng et al., 2022), and EI+our method, as risk estimate, respectively, and draw its distribution across the 140 models. Because confidence and EI leverage the sample *in isolation*, we observe spurious model responses: in (a) and (c) many models making incorrect predictions may give excessively high confidence/EI, while many making correct predictions give unexpectedly low confidence/EI. In (b) and (d), our method effectively rectifies the erroneous risk estimates, so that risk estimates of the individual sample better separate good models from poor ones. As such, the vicinal risk proxy averaged over the entire OOD test set is more indicative of model accuracy. More examples are shown in the supplementary material.

sometimes give excessively high (low) scores to incorrect (correct) samples, which compromise performance assessment; we show this problem can be alleviated by the proposed method.

**Vicinal risk** was originally introduced in the vicinal risk minimization (VRM) principle (Chapelle et al., 2000). In VRM, each training sample defines a vicinal distribution, and accordingly, model risk is evaluated based on these distributions instead of individual training samples. VRM is widely reflected in data augmentation methods (Chapelle et al., 2000; Cao and Rockett, 2015; Ni and Rockett, 2015; Zhang et al., 2017; Yun et al., 2019; Qin et al., 2020; Kim et al., 2020). For example, MixUp (Zhang et al., 2017) generates samples from vicinal distributions by mixing two images and their corresponding labels with random mixing weights. Other examples are CutMix (Yun et al., 2019), ResizeMix (Qin et al., 2020), and PuzzleMix (Kim et al., 2020). While these methods reflect vicinal risks on labeled training data, we apply this idea to unlabeled test data. Our strategy smooths out spurious proxy scores and allows for better approximation to model OOD accuracy.

## 3 PRELIMINARIES

### 3.1 RISK AND ACCURACY IN SUPERVISED EVALUATION

We consider a model $f$ belonging to a class of models $\mathcal{F}$ and a target distribution $P(\boldsymbol{x}, y)$. Model risk can be formulated as the expectation of a given loss function $\ell(f(\boldsymbol{x}), y)$ on the distribution $P$:

$$R(f) = \int \ell(f(\boldsymbol{x}), y) dP(\boldsymbol{x}, y). \tag{1}$$

In practice, since distribution $P(\boldsymbol{x}, y)$ is unknown, Eq. 1 cannot be directly computed. Standard practice thus approximates the test risk by replacing $P(\boldsymbol{x}, y)$ with an empirical distribution $P_{emp}(\boldsymbol{x}, y)$, formed by assembling Dirac delta functions (Dirac, 1981) centered at each sample in a given test set $\mathcal{D} := \{(x_i, y_i)\}_{i=1}^n$:

$$dP_{emp}(\boldsymbol{x}, y) = \frac{1}{n} \sum_{i=1}^n \delta_{\boldsymbol{x}_i}(\boldsymbol{x}) \cdot \delta_{y_i}(y). \tag{2}$$

Substituting Eq. 2 into Eq. 1, the empirical risk under empirical target distribution $P_{emp}$ becomes:

$$R_{emp}(f) = \frac{1}{n} \sum_{i=1}^n \ell(f(\boldsymbol{x}_i), y_i). \tag{3}$$

By convention, accuracy can be viewed as a type of empirical risk with the accuracy loss:

$$\ell_{acc} = \begin{cases} 0, & \text{if } \hat{y} = y, \\ 1, & \text{if } \hat{y} \neq y \end{cases},$$

where $\hat{y}$ is the predicted class with maximum (Softmax) confidence in $f(\boldsymbol{x})$.

## 3.2 VICINAL RISK MINIMIZATION

In *training*, the risk (loss values) of individual *training* samples may not fairly reflect the true generalization ability of a model on this sample. That is, the model may trivially minimize $R_{emp}(f)$ in Eq. 3 by mere memorization of the samples (Zhang et al., 2017; 2021) rather than learn effective patterns. To address this problem, *vicinal risk minimization* (Chapelle et al., 2000) suggests to replace the Dirac delta function $\delta_{\boldsymbol{x}_i}(\boldsymbol{x})$ and $\delta_{y_i}(y)$ in Eq. 2 by some density estimates of the vicinity of point $(\boldsymbol{x}_i, y_i)$:

$$dP_v(\boldsymbol{x}, y) = \frac{1}{n} \sum_{i=1}^{n} dv(\boldsymbol{x}, y | \boldsymbol{x}_i, y_i), \tag{4}$$

where $dv(\boldsymbol{x}, y | \boldsymbol{x_i}, y_i)$ is the vicinal density function describing the probability of finding point $(\boldsymbol{x}, y)$ in the vicinity of $(\boldsymbol{x}_i, y_i)$, and $P_v$ is a mixture distribution of $n$ vicinal distributions $v$. The expectation of the vicinal risk of model $f$ is now the mean of risk expectation in each vicinal distribution $v$:

$$R_v(f) = \int \ell(f(\boldsymbol{x}), y) dP_v(\boldsymbol{x}, y) = \frac{1}{n} \sum_{i=1}^{n} \int \ell(f(\boldsymbol{x}), y_i) dv(\boldsymbol{x}, y | \boldsymbol{x}_i, y_i). \tag{5}$$

The uniform vicinal distribution (Chapelle et al., 2000), Gaussian vicinal distribution (Chapelle et al., 2000) and mixup vicinal distribution (Zhang et al., 2017) are well-known vicinity in the risk minimization task (Zhang et al., 2018). The success of vinical risk minimization in *model training* (Chapelle et al., 2000; Cao and Rockett, 2015; Ni and Rockett, 2015; Hai-Yan and Hua, 2010; Dong et al., 2022; Zhang et al., 2017) demonstrates the effectiveness of this idea, which inspires us to enhance the risk estimation in the unsupervised *evaluation* problem.

## 4 PROPOSED METHOD

### 4.1 FROM RISK TO RISK PROXY IN UNSUPERVISED EVALUATION

In unsupervised evaluation, it is infeasible to use $\ell_{acc}$ defined in Eq. 3 because of the absence of ground-truths. To still be able to indicate model risk, existing methods typically design a *proxy loss* $\hat{\ell}$ and compute its expectation on the unlabeled distribution $P(\boldsymbol{x})$:

$$\widehat{R(f)} = \int \hat{\ell}(f, \boldsymbol{x}, \varphi) dP(\boldsymbol{x}), \tag{6}$$

where $\widehat{R(f)}$ is defined as *risk proxy*. $\hat{\ell}$, usually reflecting crucial model properties (*e.g.*, confidence and invariance), is computed based on the response of model $f$ to input $\boldsymbol{x}$, and additional knowledge $\varphi$ of the model. In average confidence (Tu et al., 2023) and EI (Deng et al., 2022), $\hat{\ell}$ takes the confidence of $\boldsymbol{x}$ or its transformation, so $\varphi = \emptyset$. In DoC (Guillory et al., 2021), $\varphi$ means model accuracy and average confidence evaluated on a validation set. In ATC (Garg et al., 2022), $\varphi$ is a model-specific confidence threshold learned from a validation set.

In practice, on a test set with $n$ test samples, $\widehat{R(f)}$ is approximated by the *empirical risk proxy*:

$$\widehat{R_{emp}(f)} = \frac{1}{n} \sum_{i=1}^{n} \hat{\ell}(f, \boldsymbol{x}_i, \varphi). \tag{7}$$

### 4.2 PROPOSED VICINAL RISK PROXY

**Issues of empirical risk proxies.** Existing methods in unsupervised evaluation assume that the designed risk proxy $\widehat{R(f)}$ well correlates with model generalization on the target OOD distribution. However, this assumption can be tenuous when we zoom in individual samples. As depicted in Fig. 2 (a-b), many models correctly classifying a sample have unexpected low confidence/invariance scores, and many incorrectly model predictions have excessively high confidence/invariance scores. The presence of such spurious model responses in individual samples introduces noise to the empirical

risk proxy defined in Eq. 7, rendering it less effective in assessing model generalization capabilities. Note that a dual problem that exists in *training* is described in Section 3.2.

**Solution.** Given an unlabeled test set $\mathcal{D} := \{(\boldsymbol{x}_i)\}_{i=1}^n$, we propose to compute the *vicinal risk proxy* on $\mathcal{D}$ as an unsupervised indicator of the accuracy of model $f$ on this test set. We first define vicinal distribution $\mu$ for each test sample $\boldsymbol{x}_i$ below:

$$\mu(\boldsymbol{x}, y | f, \boldsymbol{x}_i). \tag{8}$$

The probability density function for $\mu$ in this paper is defined as:

$$d\mu(\boldsymbol{x}, y | f, \boldsymbol{x}_i) = \begin{cases} s(f(\boldsymbol{x}^t), f(\boldsymbol{x}_i^t)), & \text{if } y = \widehat{y_i^t} \\ 0, & \text{if } y \neq \widehat{y_i^t} \end{cases}, \tag{9}$$

where $\boldsymbol{x}^t$ is a transformed view of $\boldsymbol{x}$, and $\widehat{y_i^t}$ is the predicted class of $\boldsymbol{x}_i^t$. There are different choices of image transformations in practice and we empirically choose rotation. $s(\cdot, \cdot)$ computes the similarity between the outputs given by model $f$, where we empirically compute the dot product between the Softmax vectors. Intuitively, $\mu$ is the probability distribution of finding pair $(\boldsymbol{x}, y)$ in the vicinity of $\boldsymbol{x}_i$, and $d\mu$ is its probability density function. Integrating such vicinal assessment into the point-wise empirical risk proxy, Eq. 7 can be updated as the vicinal risk proxy:

$$\widehat{R_v(f)} = \frac{1}{n} \sum_{i=1}^n \int \hat{\ell}(f, \boldsymbol{x}, \varphi) d\mu(\boldsymbol{x}, y | f, \boldsymbol{x}_i). \tag{10}$$

Essentially, instead of merely using $\boldsymbol{x}_i$ itself for risk estimation, we also use its neighboring samples. A sample with higher similarity with $\boldsymbol{x}_i$ contributes more to the risk. We find that spurious model responses on $\boldsymbol{x}_i$ can be effectively inhibited by its vicinal risks. For example, in Fig. 2(c-d), for a test sample, models making correct and incorrect predictions are better separated. Quantitative analysis will be provided in Section 5.

In practice, we approximate the expectation of $\hat{\ell}$ within the $i$-th distribution $\mu$ as:

$$\hat{\ell}(f, \boldsymbol{x}, \varphi) = \frac{\sum_{j=1}^m \hat{\ell}(f, \boldsymbol{x}_j, \varphi) d\mu(\boldsymbol{x}_j, \widehat{y_j} | f, \boldsymbol{x}_i)}{\sum_{j=1}^m d\mu(\boldsymbol{x}_j, \widehat{y_j} | f, \boldsymbol{x}_i)}, \tag{11}$$

where $\widehat{y_j}$ is the predicted class of $\boldsymbol{x}_j$ in $\mathcal{D}$, and $m$ is the number of samples in a vicinal distribution. Intuitively, Eq. 11 gives an empirical estimation of the risk proxy considering the vicinal distribution of $\boldsymbol{x}_i$ and the probability density defined in Eq. 9 for each vicinal sample.

For individual samples, the vicinal score allows correct and incorrect model predictions to be better separated (refer Fig. 2 for an example). Collectively on the test set, models making more correct predictions (higher accuracy) will receive higher vicinal scores than models make less correct prediction (lower accuracy). An illustrative derivation is presented in the supplementary material.

### 4.3 APPLY VICINAL RISK PROXY TO EXISTING RISK PROXIES.

In Eq. 6 and Eq. 7 of Section 4.1, we show that some existing approaches in unsupervised evaluation can be seen as unsupervised proxies for the empirical risk on the test set. Moreover, Eq. 10 means that the proposed vicinal risk proxy marries existing risk proxies with vicinal distribution. In other words, the idea of considering vicinal samples can be applied to various proxy loss functions $\hat{\ell}$. For example, when $\hat{\ell}$ is sample confidence, or equivalently, $\widehat{R_{emp}(f)}$ is the test average confidence (Tu et al., 2023; Hendrycks and Gimpel, 2017) computed empirically, also called empirical risk proxy (ERP). $\widehat{R_v(f)}$ is the the vicinal average confidence under our vicinal assessment, so is called vicinal risk proxy (VRP). By default, we search for neighboring samples for each vicinal distribution throughout the entire dataset. Samples with similarities greater than 0 are used to approximate the VRP score in Eq. 11. for this vicinity of interet.

### 4.4 DISCUSSIONS

**Why are there spurious model responses?** One possible reason for excessively high confidence for incorrect predictions is the over-confidence problem (Guo et al., 2017). This problem, when

happening in in-distribution (IND) test sets, can be rectified by model calibration, but there still lack adaptive solutions to different OOD datasets. As for excessively low confidence for correct OOD prediction, we are yet to find a convincing solution, and it would be an interesting problem to study in the future.

**Effectiveness of vicinal assessment under in-distribution test sets.** For IND data, models generally have good behaviours, so the effectiveness of vicinal assessment may be limited, but applying it does not compromise the system (partially demonstrated in Table 1). On OOD data, on the other hand, vinical assessment will be very useful as model responses are much less indicative of its correctness.

**Vicinal assessment for data-centric unsupervised evaluation.** When we assume fixed classifier and training data and vary the test data (Deng et al., 2021; Garg et al., 2022), technically vicinal assessment can still be applied. However, because it is designed to differentiate models (see Fig. 2) rather than differentiate test sets, it does not give noticeable improvement under the data-centric setup, shown in our supplementary material.

## 5 EXPERIMENTS

### 5.1 DATASETS AND EVALUATION METRICS

**ImageNet-1k setup.** 1. Model. We use 140 models that have been trained or fine-tuned using the ImageNet-1k (Deng et al., 2009) training set. We source these models from the models zoo Timm (Wightman, 2019). As suggested by Deng et al. (2022), these models exhibit a diverse range of architectures, training strategies, and pre-training settings. 2. Data. (1) *ImageNet-A(dversarial)* (Hendrycks et al., 2021b) comprises natural adversarial examples that are unmodified and occur in the real-world. (2) *ImageNet-S(ketch)* (Wang et al., 2019) contains images with a sketch-like style. (3) *ImageNet-R(endition)* (Hendrycks et al., 2021a) comprises of 30,000 images that exhibit diverse styles. (4) *ImageNet-Blur* (Hendrycks and Dietterich, 2019) was produced by applying a Gaussian function to blur the images from ImageNet-Val. We use blur with highest severity. (5) *ObjectNet* (Barbu et al., 2019) is a real-world set for object recognition with control where object backgrounds, rotations, and imaging viewpoints are random. (6) *ImageNet-V2* (Recht et al., 2019) is a reproduced ImageNet dataset, whose distribution is similar to the ImageNet dataset.

**CIFAR10 setup.** 1. Model. We use 101 models in this set sup. We follow the practice in (Deng et al., 2022) to access model weights. 2. Data. (1) *CINIC-10* (Darlow et al., 2018) is a fusion of CIFAR-10 and ImageNet-C (Hendrycks and Dietterich, 2019) image classification datasets. It contains the same 10 classes as CIFAR-10. (2) *CIFAR-10.1* (Recht et al., 2018)is produced with almost the same distribution as CIFAR-10.

**iWildCam setup.** 1. Model. We use 35 models trained on the iWildCam(Beery et al., 2020) training set. 2. Data. iWildCam-OOD test set contains animal pictures captured by camera traps in the wild.

**Evaluation metrics.** We use the same evaluation metrics as (Deng et al., 2022), *i.e.*, Pearson's Correlation coefficient ($\gamma$) (Cohen et al., 2009) and Spearman's Rank Correlation coefficient ($\rho$) (Kendall, 1948). They assess the degree of linearity and monotonicity between risk proxies and OOD accuracy, respectively. The values of both coefficients fall between -1 and 1. A coefficient being close to -1 or 1 indicates a robust negative or positive correlation. Conversely, a value of 0 denotes no correlation (Cohen et al., 2009). Following (Deng et al., 2022), we use top-1 classification accuracy as a metric to measure model generalization.

### 5.2 EXISTING RISK PROXIES AS BASELINES

We evaluate the effectiveness of vicinal assessment in enhancing the following risk proxies in unsupervised generalization prediction. 1) *Average Confidence* (AC) (Tu et al., 2023; Hendrycks and Gimpel, 2017). The mean of the softmax confidence for each samples on the test set. 2) *Effective Insurance* (EI) (Deng et al., 2022) is the multiplication between the confidence of the image and a different view of it (*e.g.*, rotation) if the predicted class of them is the same. Otherwise, it is equal to zero for this sample. 3) *Consistency Invariance* (CI) (Aithal et al., 2021) is the predicted probability of the transformed view affecting the predicted class of the original image. 4) *Difference of Confidence* (DoC) (Guillory et al., 2021) is obtained by using the accuracy on the held-out validation

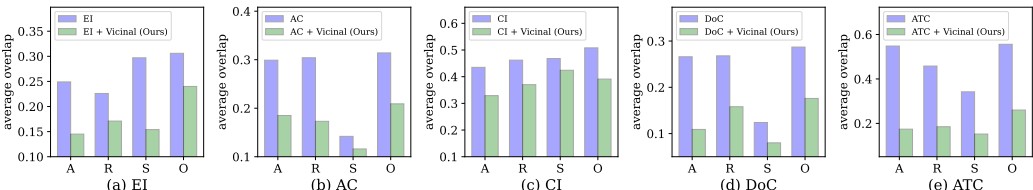

Figure 3: **Comparing existing risk estimates and their vicinal improvements.** We first estimate the distributions of risk estimate scores for correct and incorrect model predictions (140 in total) for each test sample. Then, the overlap of the two distributions for each sample is computed and finally averaged over the entire test set. All models are trained on ImageNet. In each figure, we use four test sets, ImageNet-A (A), ImageNet-R (R), ImageNet-S (S), and ObjectNet (O). From (a) to (e), EI, AC, CI, DoC, and ATC are used as baselines, respectively. *a smaller value indicates lower overlap or higher separability*. We clearly observe that vicinal risk scores statistically better differentiate models making correct and incorrect predictions by better separating their scores.

set to subtract the gap between the AC on the validation set and the AC on the test set. 5) *Average Thresholded Confidence* (ATC) (Garg et al., 2022) quantifies the proportion of samples that have a softmax confidence score exceeding the threshold learned from the validation set.

## 5.3    MAIN OBSERVATIONS

**Statistically, vicinal risk scores better differentiate correct and incorrect model predictions.** Apart from the example shown in Fig. 2, we further provide statistical evidence of the working mechanism of our method. Specifically, we employ the Gaussian kernel density estimation (KDE) to estimate the distributions of proxy scores of models that make correct and incorrect predictions for each sample. Then, we use numerical integration to calculate the their overlap, known as the overlap coefficient. In Fig. 3, we present the average coefficient across each test set. Results demonstrate that vicinal assessment has lower overlap coefficient than the baseline proxies, making it easier to differentiate between correct and incorrect predictions at the individual sample level.

**Vicinal assessment consistently improves existing risk proxies on OOD test sets.** In Table 1, we compare five existing risk proxies and their vicinal versions on OOD test sets. Each experiment is repeated three times to show statistical significance. We observe a consistent improvement in the strength of correlation between accuracy and risk proxy. For example, on ImageNet-A, vicinal assessment brings about 4.8%, 3.9%, 10.5%, 3.6%, and 4.5% improvement in the Spearman's coefficient over EI, AC, CI, DoC, and ATC, respectively. These results indicate the effectiveness of the proposed method. The improvements can be illustrated by two examples in Fig, 4, where the ranks of the majority models have been adjusted to be closer to the actual rank.

**Vicinal assessment is neither beneficial nor detrimental on near-OOD test sets.** When test data are near OOD or even IND, the use of vicinal assessment does not have noticeable performance improvement or compromise. For example, on ImageNet-V2, we observe slight improvement for EI, CI, DoC and ATC, and slight decrease for AC. On the CIFAR10.1 test set, similar observations are made. Further given its effectiveness in OOD scenarios, this allows for safe deployment of vicinal assessment in practice.

## 5.4    FURTHER ANALYSIS OF VICINAL RISK PROXY

**Comparing different similarity measurements in Eq. 9**. Dot product is used in Eq. 9 to compute similarity between a sample of interest and a sample in its vicinity. Here, we compare it with other options including random value (giving random similarity values), equal similarity (*i.e.*, uniform vicinal distribution (Chapelle et al., 2000)), and the Gaussian kernel function (*i.e.*, Gaussian vicinal distribution (Chapelle et al., 2000)). Results of ranking the 140 models on ImageNet-A are summarized in Table 2. We find that dot product generally has similar performance with Gaussian similarity, and both are much better than random similarity and equal similarity. It illustrates the benefit of letting closer sample to contribute more to the score of the sample of interest.

Table 1: Comparing vicinal risk proxies (VRP) and empirical risk proxies (ERP) on various test sets. For each test set, results in the first row is from ERP and the second is VRP. $\gamma$ and $\rho$ represent the Pearson's coefficient and the Spearman's correlation coefficient, respectively. ♣ means near IND test sets. The notations ↑ (↓) on our score means the correlation coefficient of VRP is higher (lower) than that of ERP with statistical significance (p-value < 0.05) based on the two-sample t-test. Otherwise, their difference is not statistical significant.

| Train | Test | EI | | AC | | CI | | DoC | | ATC | |
|---|---|---|---|---|---|---|---|---|---|---|---|
| | | $\gamma$ | $\rho$ | $\gamma$ | $\rho$ | $\gamma$ | $\rho$ | $\gamma$ | $\rho$ | $\gamma$ | $\rho$ |
| ImageNet | ImageNet-A | 0.882 | 0.645 | 0.581 | 0.464 | 0.856 | 0.617 | 0.877 | 0.761 | 0.851 | 0.436 |
| | | 0.900↑ | 0.692↑ | 0.624↑ | 0.503↑ | 0.905↑ | 0.722↑ | 0.908↑ | 0.797↑ | 0.866↑ | 0.481↑ |
| | ImageNet-R | 0.914 | 0.814 | 0.736 | 0.625 | 0.873 | 0.729 | 0.898 | 0.862 | 0.937 | 0.887 |
| | | 0.956↑ | 0.931↑ | 0.818↑ | 0.736↑ | 0.931↑ | 0.854↑ | 0.905↑ | 0.894↑ | 0.967↑ | 0.946↑ |
| | ImageNet-S | 0.893 | 0.853 | 0.742 | 0.711 | 0.868 | 0.820 | 0.911 | 0.919 | 0.948 | 0.915 |
| | | 0.920↑ | 0.871↑ | 0.763↑ | 0.728↑ | 0.878↑ | 0.840↑ | 0.926↑ | 0.931↑ | 0.954↑ | 0.953↑ |
| | ObjectNet | 0.961 | 0.949 | 0.788 | 0.777 | 0.958 | 0.946 | 0.819 | 0.834 | 0.841 | 0.860 |
| | | 0.975↑ | 0.972↑ | 0.838↑ | 0.814↑ | 0.969↑ | 0.962↑ | 0.849↑ | 0.868↑ | 0.857↑ | 0.876↑ |
| | ImageNet-Blur | 0.870 | 0.831 | 0.711 | 0.730 | 0.824 | 0.793 | 0.781 | 0.776 | 0.882 | 0.867 |
| | | 0.907↑ | 0.857↑ | 0.737↑ | 0.741↑ | 0.829↑ | 0.802↑ | 0.821↑ | 0.821↑ | 0.912↑ | 0.890↑ |
| | ImageNet-V2♣ | 0.889 | 0.884 | 0.609 | 0.501 | 0.882 | 0.870 | 0.982 | 0.979 | 0.993 | 0.990 |
| | | 0.895↑ | 0.881 | 0.613 | 0.513 | 0.886↑ | 0.887↑ | 0.990↑ | 0.984↑ | 0.995 | 0.993 |
| CIFAR10 | CINIC | 0.913 | 0.936 | 0.978 | 0.887 | 0.834 | 0.876 | 0.985 | 0.953 | 0.983 | 0.937 |
| | | 0.954↑ | 0.956↑ | 0.979 | 0.889↑ | 0.875↑ | 0.929 | 0.985 | 0.956↑ | 0.982 | 0.942↑ |
| | CIFAR10.1♣ | 0.886 | 0.905 | 0.982 | 0.972 | 0.804 | 0.811 | 0.992 | 0.985 | 0.991 | 0.982 |
| | | 0.883 | 0.886↓ | 0.982 | 0.972 | 0.813↑ | 0.855↑ | 0.992 | 0.985 | 0.991 | 0.982 |
| iWildCam | iWildCam-OOD | 0.337 | 0.362 | 0.635 | 0.445 | 0.268 | 0.258 | 0.547 | 0.532 | 0.509 | 0.526 |
| | | 0.402↑ | 0.393↑ | 0.655↑ | 0.495↑ | 0.208↓ | 0.180↓ | 0.556↑ | 0.592↑ | 0.518↑ | 0.595↑ |

(a) ImageNet-R                                   (b) ObjectNet

Figure 4: **Correlation between Effective Invariance (EI) and accuracy**. In each figure, every dot represents a model, and straight lines are fitted using a robust linear fit. A blue dots represent the rectified score (VRP score) of this models brings its rank closer to the actual accuracy rank. On the other hand, the rank of red models deviates further from the real accuracy when using the VRP paradigm. The rank of black models remains unchanged. $\rho$ and $\gamma$ have the same meaning as those in Table 1 The shaded region in each figure represents a 95% confidence region for the linear fit, calculated from 1,000 bootstrap samples. We observe that the VRP paradigm can effectively rectify the proxy score for the majority of models in both the ImageNet-R and ObjectNet datasets.

**Comparing different image transformations.** In Eq. 9, we define the probability density function for vicinal distribution using transformed images. Here we compare the effectiveness of different transformations, and results on ImageNet-A are presented in Table 2. We find that there is no significant performance difference between rotation, gray-scale transformation and color jitters.

**Impact of the number of neighbors.** The number of neighbors $m$ is an important hyper-parameter used in Eq. 11. To evaluate system sensitivity to $m$, we experiment with ImageNet-R as OOD test set and five baseline risk proxies setting $m = 25, 50, 75, 100, 125, 150$[1]. We repeated each experiment three times and reported the mean and standard deviation. From Fig. 5 we have two observations.

---

[1]Because of the limited size of ImageNet-R, 150 is the maximum value we can set $m$ to.

Table 2: **Comparison of variants of vicinal risk proxy. (Top)**: various similarity metrics that can be used in Eq. 9. **(Bottom)**: various image transformations that can be used in Eq. 9. Bold numbers denote the best one across compared settings.

| Settings | EI | | AC | | CI | | DoC | | ATC | |
|---|---|---|---|---|---|---|---|---|---|---|
| | $\gamma$ | $\rho$ | $\gamma$ | $\rho$ | $\gamma$ | $\rho$ | $\gamma$ | $\rho$ | $\gamma$ | $\rho$ |
| Random | 0.003 | 0.197 | 0.072 | 0.133 | 0.216 | 0.350 | 0.073 | 0.160 | 0.066 | 0.055 |
| Equal | 0.883 | 0.659 | 0.554 | 0.446 | 0.857 | 0.624 | 0.864 | 0.757 | 0.837 | 0.424 |
| Gaussian kernel | 0.876 | 0.675 | **0.611** | **0.498** | 0.887 | 0.706 | 0.887 | 0.797 | **0.881** | **0.532** |
| Dot product | **0.903** | **0.713** | 0.605 | 0.489 | **0.903** | **0.731** | **0.901** | **0.801** | 0.867 | 0.490 |
| None | 0.901 | 0.705 | 0.564 | 0.480 | 0.881 | 0.669 | 0.907 | 0.806 | 0.863 | 0.487 |
| Grey-scale | 0.898 | 0.686 | **0.617** | **0.503** | 0.879 | 0.655 | 0.917 | **0.812** | **0.889** | **0.514** |
| Color jitters | 0.897 | 0.674 | 0.632 | 0.512 | 0.861 | 0.642 | **0.919** | 0.811 | 0.874 | 0.482 |
| Rotation | **0.903** | **0.713** | 0.605 | 0.489 | **0.903** | **0.731** | 0.901 | 0.801 | 0.867 | 0.490 |

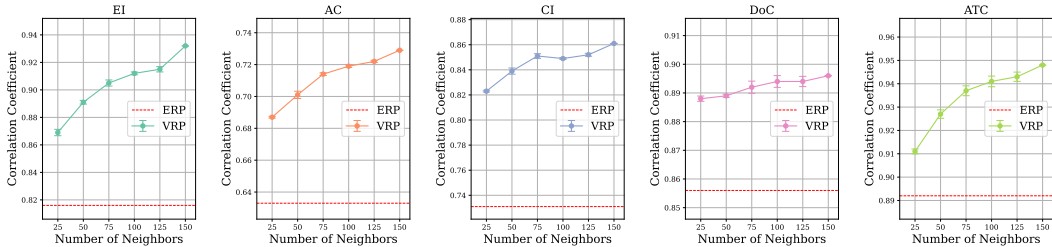

Figure 5: **Impact of the number of neighbors $m$ on the correlation between proxy scores and accuracy.** We use five existing proxies as baselines and report the mean and standard deviation for each data point. We observe that vicinal assessment is consistently beneficial under various $m$ values and yields stronger correlation with $m$ increases.

First, using more neighboring samples is generally beneficial, evidenced by the increasing correlation strength. It is probably because a larger $m$ allows for better approximation of the true vicinal distribution. We set $m = 150$ by default. Second, when using fewer neighbors, *e.g.,* $m = 25$, vicinal assessment is still beneficial, yielding stronger correlation than existing risk proxies.

**Impact of test set size.** Table 3 presents our method under varying test set sizes using ImageNet-R as an OOD test set. We observe that the performance of all compared methods drops under smaller test sets. Nevertheless, the use of vicinal assessment consistently improves the correlation strength of the baselines under each test set size, demonstrating the effectiveness of our method.

Table 3: **Impact of the test set size.** We evaluate our method on the ImageNet-R set with different number of test samples.

| # samples | 3,000 | 6,000 | 12,000 | 18,000 | 24,000 |
|---|---|---|---|---|---|
| DoC | 0.856 | 0.848 | 0.877 | 0.874 | 0.874 |
| DoC + Ours | **0.873** | **0.874** | **0.892** | **0.895** | **0.896** |

## 6 CONCLUSION

In this paper, we propose the vicinal assessment strategy to improve existing risk proxies computed based on a single test sample. We demonstrate that existing point-wise methods are prone to erroneous model responses, a problem that can be alleviated by considering the responses of adjacent test samples. Inspired by the philosophy of vicinal risk minimization, we design a vicinal risk proxy. We find that its computation on individual samples better differentiates models that make correct predictions from those that make incorrect ones. Therefore, when averaged across the test set, the vicinal risk proxy more accurately reflects the out-of-distribution (OOD) generalization ability of models. This main conclusion is verified through extensive experiments and further supported by analysis of its variants, sensitivity to key hyper-parameters, and application scope.

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
