# VICINAL ASSESSMENT OF MODEL GENERALIZATION

## A    EXPERIMENTAL SETUP

In this section, we introduce how to access the models and datasets used in our paper.

### A.1    IMAGENET SETUP

**Models.** Following the practice of Deng et al. (2022), we use the ImageNet models provided by PyTorch Image Models (timm) (Wightman, 2019). It provides models trained or fine-tuned on the ImageNet-1k training set (Deng et al., 2009). We show the names of models used in our paper below:

'esmlp_36_224', 'cait_s36_384', 'cait_s24_224', 'convit_base', 'convit_tiny', 'twins_pcpvt_base', 'eca_nfnet_l1', 'xcit_tiny_24_p8_384_dist', 'efficientnet_b1', 'efficientnet_b3' 'efficientnet_b4', 'tf_efficientnet_b2', 'tf_efficientnet_lite1', 'convnext_base', 'convnext_small', 'resnetrs350', 'pit_xs_distilled_224', 'crossvit_small_240', 'botnet26t_256', 'tinynet_e', 'tinynet_d', 'repvgg_b2g4', 'mnasnet_small', 'dla46x_c', 'lcnet_050', 'tv_resnet34', 'tv_resnet50', 'tv_resnet101' 'tv_resnet152' 'densenet121' 'inception_v4' 'resnet26d' 'mobilenetv2_140', 'hrnet_w40', 'xception', 'xception41', 'resnet18', 'resnet34', 'seresnet50', 'mobilenetv2_050', 'seresnet33ts', 'wide_resnet50_2', 'wide_resnet101_2', 'resnet18d', 'hrnet_w18_small', 'gluon_resnet152_v1d', 'hrnet_w48', 'hrnet_w44', 'repvgg_b2', 'densenet201', 'hrnet_w18_small', 'resnet101d', 'gluon_resnet101_v1d', 'gluon_resnet101_v1s', 'gluon_xception65', 'gluon_seresnext50_32x4d', 'gluon_senet154', 'gluon_inception_v3', 'gluon_resnet101_v1c', 'tf_inception_v3', 'tv_densenet121', 'tv_resnext50_32x4d', 'repvgg_b1g4', 'resnext26ts', 'ghostnet_100', 'crossvit_9_240', 'deit_base_patch16_384', 'rexnet_150', 'rexnet_130', 'resnetrs50', 'resnet50d', 'resnet50', 'resnetv2_50', 'resnetrs152', 'resnetrs101', 'dpn92', 'dpn98', 'dpn68', 'vgg19_bn', 'vgg16_bn', 'vgg13_bn', 'vgg11_bn', 'vgg11', 'vgg11_bn', 'vgg16', 'vgg19', 'swin_small_patch4_window7_224', 'swin_base_patch4_window12_384', 'deit_base_patch16_224', 'deit_small_distilled_patch16_224', 'densenet161', 'tf_mobilenetv3_large_075', 'inception_v3', 'ssl_resnext101_32x8d', 'ssl_resnext101_32x16d', 'swsl_resnext101_32x8d', 'swsl_resnext101_32x16d', 'ssl_resnext101_32x4d', 'ssl_resnext50_32x4d', 'ssl_resnet50', 'swsl_resnext101_32x4d', 'swsl_resnext50_32x4d', 'swsl_resnet50', 'tf_efficientnet_l2_ns_475', 'tf_efficientnet_b7_ns', 'tf_efficientnet_b6_ns', 'tf_efficientnet_b4_ns', 'tf_efficientnet_b5_ns', 'convnext_xlarge_384_in22ft1k', 'convnext_xlarge_in22ft1k', 'convnext_large_384_in22ft1k', 'convnext_large_in22ft1k', 'convnext_base_384_in22ft1k', 'convnext_base_in22ft1k', 'resnetv2_152x2_bitm', 'resnetv2_152x4_bitm', 'resnetv2_50x1_bitm', 'resmlp_big_24_224_in22ft1k', 'resmlp_big_24_distilled_224', 'tf_efficientnetv2_s_in21ft1k', 'tf_efficientnetv2_m_in21ft1k', 'tf_efficientnetv2_l_in21ft1k', 'tf_efficientnetv2_xl_in21ft1k', 'vit_large_patch16_384', 'swin_large_patch4_window12_384', 'beit_large_patch16_512', 'beit_large_patch16_384', 'beit_large_patch16_224', 'beit_base_patch16_384', 'vit_base_patch16_384', 'vit_small_r26_s32_384', 'vit_tiny_patch16_384', 'vit_large_r50_s32_384', 'mixer_b16_224_miil' 'resmlp_big_24_224', 'resnetv2_50x1_bit_distilled', 'ig_resnext101_32x16d', 'ig_resnext101_32x32d', 'ig_resnext101_32x8d', 'ig_resnext101_32x48d', 'regnety_016', 'regnety_032'.

**Datasets.** We present the test sets employed in the main paper to evaluate the aforementioned ImageNet models. Datasets mentioned below can be accessed publicly via the provided links.

*ImageNet-A(dversarial)* (Hendrycks et al., 2021b) : https://github.com/hendrycks/natural-adv-examples.
*ImageNet-R(endition)* (Hendrycks et al., 2021a): https://github.com/hendrycks/imagenet-r.
*ImageNet-Blur* (Hendrycks and Dietterich, 2019) : https://github.com/hendrycks/robustness.
*ImageNet-S(ketch)* (Wang et al., 2019) : https://github.com/HaohanWang/ImageNet-Sketch.

*ImageNet-V2* (Recht et al., 2019) : https://github.com/modestyachts/ImageNetV2.
*ObjectNet* (Barbu et al., 2019) : https://objectnet.dev/download.html.

## A.2 CIFAR10 SETUP

**Models.** We employ 101 models trained on the CIFAR10 training set. Among them, 82 are trained based on the implementation from https://github.com/kuangliu/pytorch-cifar, following the practice of Deng et al. (2022). These models vary in their architectures and number of training epochs. Specifically, the following architectures are used:

*'DenseNet121', 'DenseNet169', 'DenseNet201', 'DenseNet161', 'densenet_cifar', 'DLA', 'SimpleDLA', 'DPN26', 'DPN92', 'EfficientNetB0', 'GoogLeNet', 'LeNet', 'MobileNet', 'MobileNetV2', 'PNASNetA', 'PNASNetB', 'PreActResNet18', 'PreActResNet34', 'PreActResNet50', 'PreActResNet101', 'PreActResNet152', 'RegNetX_200MF', 'RegNetX_400MF', 'RegNetY_400MF', 'ResNet18', 'ResNet34', 'ResNet50', 'ResNet101', 'ResNet152', 'ResNeXt29_2x64d', 'ResNeXt29_4x64d', 'ResNeXt29_8x64d', 'ResNeXt29_32x4d', 'SENet18', 'ShuffleNetG2', 'ShuffleNetG3', 'ShuffleNetV2', 'VGG11', 'VGG13', 'VGG16', 'VGG19'.*

For each architecture, we train two variants with 30 and 50 training epochs, respectively.

The rest 19 models are download from https://github.com/chenyaofo/pytorch-cifar-models. Names of these models are listed below:

*'cifar10_mobilenetv2_x0_5', 'cifar10_mobilenetv2_x0_75', 'cifar10_mobilenetv2_x1_0', 'cifar10_mobilenetv2_x1_4', 'cifar10_repvgg_a0', 'cifar10_repvgg_a1', 'cifar10_repvgg_a2', 'cifar10_resnet20', 'cifar10_resnet32', 'cifar10_resnet44', 'cifar10_resnet56', 'cifar10_shufflenetv2_x0_5', 'cifar10_shufflenetv2_x1_0', 'cifar10_shufflenetv2_x1_5', 'cifar10_shufflenetv2_x2_0', 'cifar10_vgg11_bn', 'cifar10_vgg13_bn', 'cifar10_vgg16_bn', 'cifar10_vgg19_bn'.*

**Datasets.** Datasets used in the CIFAR10 setup can be found through the following links.

*CIFAR10* (Krizhevsky et al., 2009) : https://www.cs.toronto.edu/ kriz/cifar.html.
*CIFAR10.1* (Recht et al., 2018) : https://github.com/modestyachts/CIFAR-10.1.
*CINIC* (Darlow et al., 2018) : https://github.com/BayesWatch/cinic-10.

## A.3 IWILDCAM SETUP

**Models.** We access 34 models trained on the iWildCam training set from the official implementation of the iWildCam benchmark (https://worksheets.codalab.org/worksheets/0x52cea 64d1d3f4fa89de326b4e31aa50a). All models use Resnet50 (He et al., 2016) as the backbone but differ in training algorithms, learning rate, weight decay, *etc*. Their identification names are provided below.

*'iwildcam_afn_extraunlabeled_tune0', 'iwildcam_dann_coarse_extraunlabeled_tune0', 'iwildcam_deepcoral_coarse_extraunlabeled_tune0', 'iwildcam_deepcoral_coarse_singlepass_extraunlabeled_tune0', 'iwildcam_deepCORAL_seed0', 'iwildcam_deepCORAL_seed1', 'iwildcam_deepCORAL_seed2' 'iwildcam_deepCORAL_tune', 'iwildcam_ermaugment_tune0', 'iwildcam_ermoracle_extraunlabeled_tune0', 'iwildcam_erm_seed0', 'iwildcam_erm_seed1', 'iwildcam_erm_seed2', 'iwildcam_erm_tune0', 'iwildcam_erm_tuneA_seed0', 'iwildcam_erm_tuneB_seed0', 'iwildcam_erm_tuneC_seed0', 'iwildcam_erm_tuneD_seed0', 'iwildcam_erm_tuneE_seed0', 'iwildcam_erm_tuneF_seed0', 'iwildcam_erm_tuneG_seed0', 'iwildcam_erm_tuneH_seed0', 'iwildcam_fixmatch_extraunlabeled_tune0', 'iwildcam_groupDRO_seed0', 'iwildcam_groupDRO_seed1', 'iwildcam_groupDRO_seed2', 'iwildcam_irm_seed0', 'iwildcam_irm_seed1', 'iwildcam_irm_seed2', 'iwildcam_irm_tune', 'iwildcam_noisystudent_extraunlabeled_seed0', 'iwildcam_pseudolabel_extraunlabeled_tune0', 'iwildcam_swav30_ermaugment_seed0'.*

**Dataset.** *iWildCam-OOD* (Beery et al., 2020) can be download from the the official guidance: https://github.com/p-lambda/wilds/.

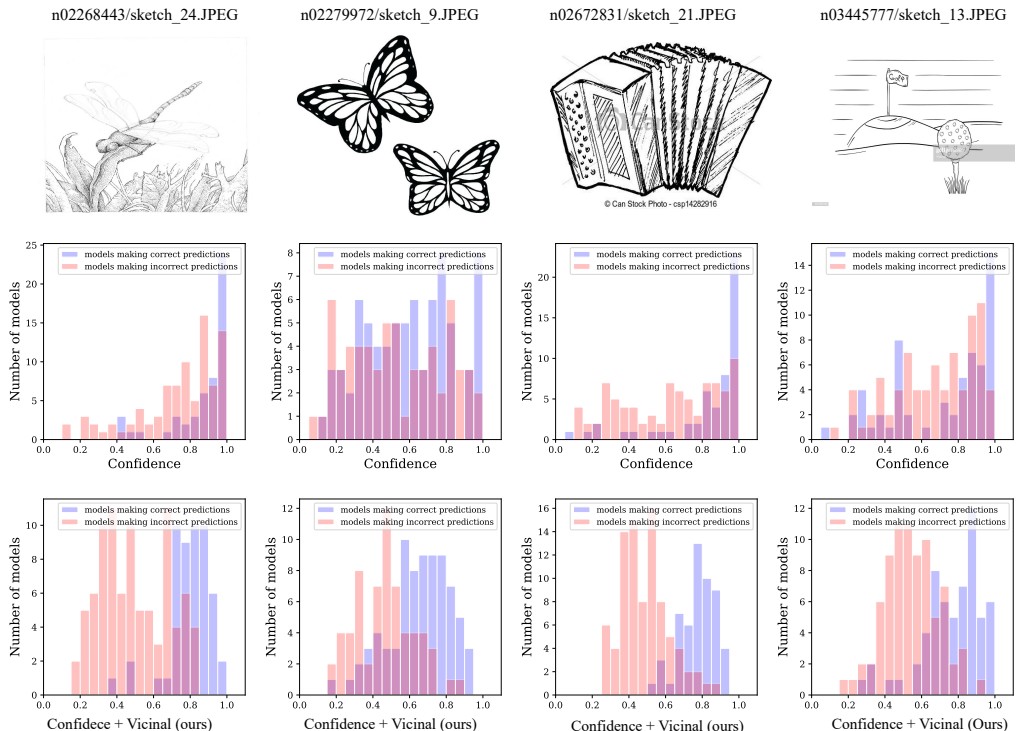

Figure A1: **Illustration of working mechanism of vicinal assessment on individual samples from ImageNet-S.** Four test samples are shown in the top row. The AC method is used as baseline proxy. Score distribution of 140 models trained on the ImageNet training set are drawn below. Notations have the same meaning as Fig. 2 in the main paper.

## B    WORKING MECHANISM ON INDIVIDUAL TEST SAMPLES

As discussed in the working mechanism part (see Section 4.2 of the main paper), we attribute the effectiveness of our method to its ability to distinguish correct and incorrect model predictions on an individual sample level, and thus better separability of models with different OOD performance. In addition to the example shown in Fig. 2 of the main paper, we provide more examples to demonstrate the working mechanism on individual test samples in Fig. A1, Fig. A2, and Fig A3. We clearly observe that the distributions of model risk estimates of incorrect and correct predictions are generally more spreadable by using proposed vicinal assessment. These examples further showcase the working mechanism, where samples in the vicinity effectively rectify erroneous risk estimates, so that risk estimates of individual samples better differentiate models making corrent and incorrect predictions.

## C    WORKING MECHANISM ON A WHOLE DATASET

**Definitions.** There are two models, $f_a$, and $f_b$, to test. We denote each test sample in the test set as $\boldsymbol{x}_i, i \in \{1, 2, \ldots, n\}$. Given an baseline empirical risk proxy $\widehat{R_e}$ (*e.g.*, AC, EI, and DoC), the empirical risk score of model $f_a$ on the test sample $\boldsymbol{x}_i$ is written as $\widehat{R_e}(f_a, \boldsymbol{x}_i)$. Similarly, the vicinal risk score of $f_a$ in the vicinity of $\boldsymbol{x}_i$ can be written as $\widehat{R_v}(f_a, \boldsymbol{x}_i)$. We assume $f_a$ has higher accuracy than $f_b$ and the risk proxy score is positively related to the model accuracy.

Here, We define $p_e^i$ as the probability of the event that the empirical risk score for $f_a$ on the test sample $\boldsymbol{x}_i$ is higher than that for $f_b$, where

$$p_e^i = P((\widehat{R_e}(f_a, \boldsymbol{x}_i) - \widehat{R_e}(f_b, \boldsymbol{x}_i)) > 0).$$

Similarly, we have

$$p_v^i = P((\widehat{R_v}(f_a, \boldsymbol{x}_i) - \widehat{R_v}(f_b, \boldsymbol{x}_i)) > 0).$$

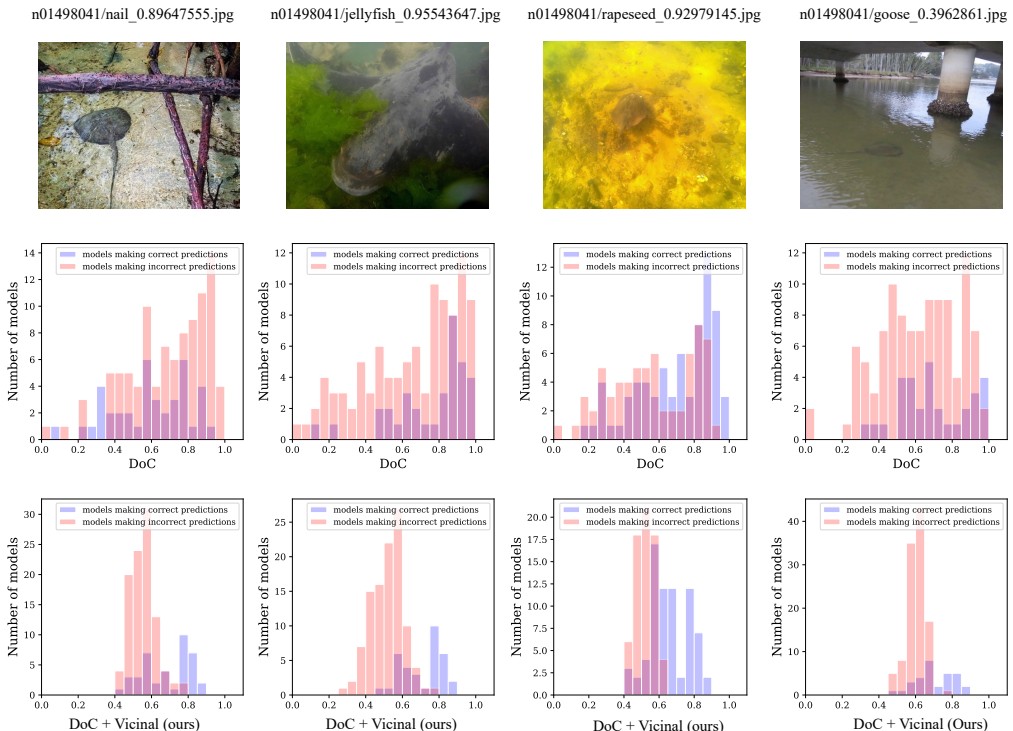

Figure A2: **Illustration of working mechanism of vicinal assessment on individual samples from ImageNet-A.** DoC is used as baseline proxy. Other notations have the same meaning as Fig. A1.

**Working mechanism.** Given the preliminary assumption that $\widehat{R_v}$ enables the score distributions of correct and incorrect model predictions for each test sample to be more separable (shown by our experiments in Fig. 3 of the main paper), there are some arbitrary samples $\{\boldsymbol{x}_i\}, i \in \{1, 2, \ldots, m\}, m \leq n$, where

$$(\frac{1}{m}\sum_i^m(p_v^i - p_e^i)) > 0 \Rightarrow (\frac{1}{m}\sum_i^m p_v^i - \frac{1}{m}\sum_i^m p_e^i) > 0.$$

Therefore, we have the following probability inequality:

$$P((\frac{1}{m}\sum_i^m\widehat{R_v}(f_a, \boldsymbol{x}_i) - \frac{1}{m}\sum_i^m\widehat{R_v}(f_b, \boldsymbol{x}_i)) > 0) > P((\frac{1}{m}\sum_i^m\widehat{R_e}(f_a, \boldsymbol{x}_i) - \frac{1}{m}\sum_i^m\widehat{R_e}(f_b, \boldsymbol{x}_i)) > 0).$$

According to Eq. 10 in the main paper, the above inequality can be rewritten as

$$P((\widehat{R_v(f_a)} - \widehat{R_v(f_b)}) > 0) > P((\widehat{R_e(f_a)} - \widehat{R_e(f_b)}) > 0),$$

where $R_v(f_a)$ and $R_e(f_a)$ is the vicinal score and the empirical risk score, respectively. It means that our method has the higher probability to successfully rank models.

# D  TIME COMPLEXITY

The computational complexity of our method is $O(nm)$, where $n$ is test set size, and $m$ is the number of neighboring samples used for VRP computation. In our algorithm, neighoring samples are those sharing the same predicted label as the sample of interest, so looking for neighboring samples does not require a search process. In our implementation, $m$ is a hyperparameter that can be as small as 25 or 50 (see Fig. 6) to yield improvement, while a test sample may have 150 neighbors. So the computational complexity is much less than $O(n^2)$.

We would like to present the experimental results of time consumption under different dataset sizes here. In Table D1, we provide the running time of evaluating the model 'vit_large_r50_s32_384' on the ImageNet-R dataset. We used NVIDIA28 V100 with $4 \times$ GPU.

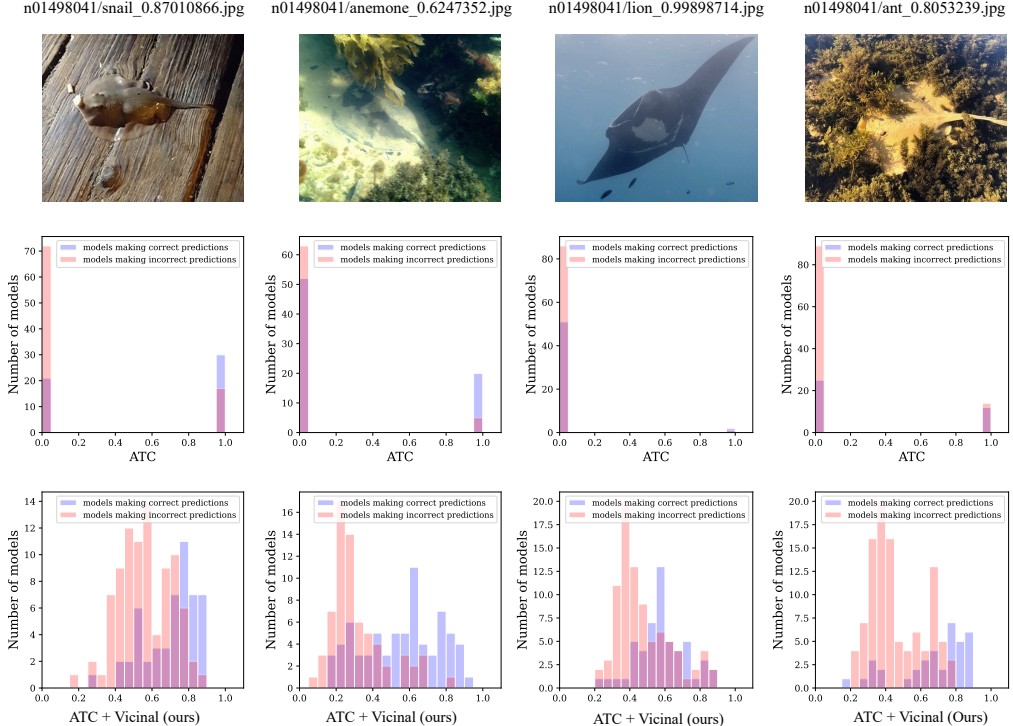

Figure A3: **Illustration of working mechanism of vicinal assessment on individual samples from ImageNet-R.** ATC is used as baseline proxy. Other notations have the same meaning as Fig. A1.

Table D1: Running time (seconds) vs. test set size for average confidence (AC) and our method (AC+VRP). The last row presents the runtime increase caused by applying VRP to AC.

| Test set size | 5,000 | 10,000 | 15,000 | 20,000 | 25,000 | 30,000 |
|---|---|---|---|---|---|---|
| AC (sec.) | 1.75 | 1.88 | 1.95 | 1.99 | 2.14 | 2.22 |
| AC + VRP (sec.) | 4.38 | 4.57 | 4.76 | 5.02 | 5.23 | 5.40 |
| Added time (sec.) | 2.63 | 2.69 | 2.81 | 3.03 | 3.09 | 3.18 |

Two main observations can be drawn from the above table. First, compared with the baseline average confidence (AC) method, applying VRP (AC+VRP) consumes an additional 2-4 seconds when the test set size is 30k.

Second, the runtime of VRP increases almost linearly with the test set size. For example, VRP runtime increases by 1.02 seconds and when the test set size increases from 5k to 30k.

## E    MORE EXPERIMENTS ON VARIANTS OF VICINAL RISK PROXY

We conduct experiments using different variants of the vicinal risk proxy on the ImageNet-R and ObjectNet datasets. The experimental results in Table E2 show similar observations to those in the main manuscript.

## F    VICINAL ASSESSMENT FOR DATA-CENTRIC UNSUPERVISED EVALUATION

As discussed in Section 4.4 of the main paper, vicinal assessment technically can be applied in data-centric unsupervised evaluation by measuring scores of a fixed model on various test sets. To evaluate this, we conduct experiments using two models, *Efficientet-b2* and *Inception-v4*, obtained

Table E2: **Comparison of variants of vicinal risk proxy on ImageNet-R (Top) and ObjectNet (Bottom).** Notations follow Tab. 2 of the main submission.

| Settings | EI | | AC | | CI | | DoC | | ATC | |
|---|---|---|---|---|---|---|---|---|---|---|
| | $\gamma$ | $\rho$ | $\gamma$ | $\rho$ | $\gamma$ | $\rho$ | $\gamma$ | $\rho$ | $\gamma$ | $\rho$ |
| Random (to do) | 0.335 | 0.399 | 0.277 | 0.331 | 0.415 | 0.466 | 0.237 | 0.461 | 0.357 | 0.563 |
| Equal | 0.910 | 0.816 | 0.738 | 0.633 | 0.868 | 0.731 | 0.897 | 0.856 | 0.935 | 0.892 |
| Dot product | 0.955 | 0.932 | 0.817 | 0.729 | 0.925 | 0.861 | 0.909 | 0.896 | 0.966 | 0.948 |
| None | 0.816 | 0.743 | 0.704 | 0.606 | 0.874 | 0.731 | 0.889 | 0.825 | 0.894 | 0.786 |
| Grey-scale | 0.945 | 0.922 | 0.816 | **0.756** | **0.944** | **0.891** | 0.901 | **0.901** | **0.971** | **0.951** |
| Color jitters | 0.930 | 0.899 | 0.813 | 0.748 | 0.941 | 0.889 | 0.880 | 0.886 | 0.969 | 0.947 |
| Rotation | **0.955** | **0.932** | **0.817** | 0.729 | 0.925 | 0.861 | **0.909** | 0.896 | 0.966 | 0.948 |
| Random | 0.024 | 0.296 | 0.165 | 0.235 | 0.306 | 0.451 | 0.133 | 0.255 | 0.172 | 0.189 |
| Equal | 0.894 | 0.851 | 0.748 | 0.714 | 0.865 | 0.819 | 0.913 | 0.921 | 0.950 | 0951 |
| Dot product | **0.917** | **0.871** | **0.763** | **0.729** | **0.875** | **0.837** | **0.927** | **0.932** | **0.955** | **0.955** |
| None | 0.963 | 0.959 | 0.807 | 0.786 | 0.965 | 0.953 | **0.854** | **0.875** | 0.825 | 0.860 |
| Grey-scale | 0.958 | 0.965 | 0.835 | **0.819** | 0.955 | 0.939 | 0.845 | 0.851 | 0.859 | 0.876 |
| Color jitters | 0.955 | 0.937 | 0.833 | 0.804 | 0.948 | 0.924 | 0.844 | 0.862 | **0.864** | **0.880** |
| Rotation | **0.975** | **0.972** | **0.838** | 0.814 | **0.969** | **0.963** | 0.849 | 0.868 | 0.857 | 0.876 |

Table F3: **Effectiveness of vicinal assessment in data-centric unsupervised evaluation.** For each model, the *first* row shows results of baseline risk proxies, while the *second* row gives results of their vicinal assessments. $\gamma$ and $\rho$ have the same meaning as described in the main paper.

| Model | EI | | AC | | CI | | DoC | | ATC | |
|---|---|---|---|---|---|---|---|---|---|---|
| | $\gamma$ | $\rho$ | $\gamma$ | $\rho$ | $\gamma$ | $\rho$ | $\gamma$ | $\rho$ | $\gamma$ | $\rho$ |
| Efficientnet-b2 | 0.932 | 0.976 | 0.987 | 0.993 | 0.919 | 0.965 | 0.987 | 0.992 | 0.933 | 0.939 |
| | 0.942 | 0.979 | 0.989 | 0.994 | 0.918 | 0.961 | 0.989 | 0.994 | 0.937 | 0.938 |
| Inception-V4 | 0.938 | 0.970 | 0.960 | 0.988 | 0.931 | 0.963 | 0.960 | 0.988 | 0.991 | 0.994 |
| | 0.942 | 0.973 | 0.964 | 0.990 | 0.928 | 0.960 | 0.964 | 0.990 | 0.993 | 0.995 |

from the model zoo in Section A.1, on 95 testing sets sourced from the test set pool, ImageNet-C (Hendrycks and Dietterich, 2019). The experimental results, showcasing the performance the vicinal method under different risk proxies, are presented in Table F3.

Our main observation is that the correlation results obtained using vicinal samples are similar to those relying merely on individual samples. In other words, there is no noticeable improvements as opposed to those model-centric experiments in the main paper. In fact, as illustrated in Fig. 1-4, vicinal proxies are helpful for distinguishing models *w.r.t* their OOD accuracy. Its limited capability in distinguishing hard datasets from easy ones leads to the observations in Table F3.

## G    MORE VISUALIZATIONS OF IMPROVED CORRELATIONS

In Fig. 5 of the main manuscript, we visualized correlations between effective invariance (EI) and accuracy and the improvement brought by vicinal proxies on the ImgeNet-R and ObjectNet datasets. Here, we present more visualizations, using AC, CI, DOC and ATC proxies. Results are shown in Fig. F4, Fig. F5 and Fig. F6. We observe that generally the proposed vicinal assessment allows more models to get closer to the actual accuracy rank.

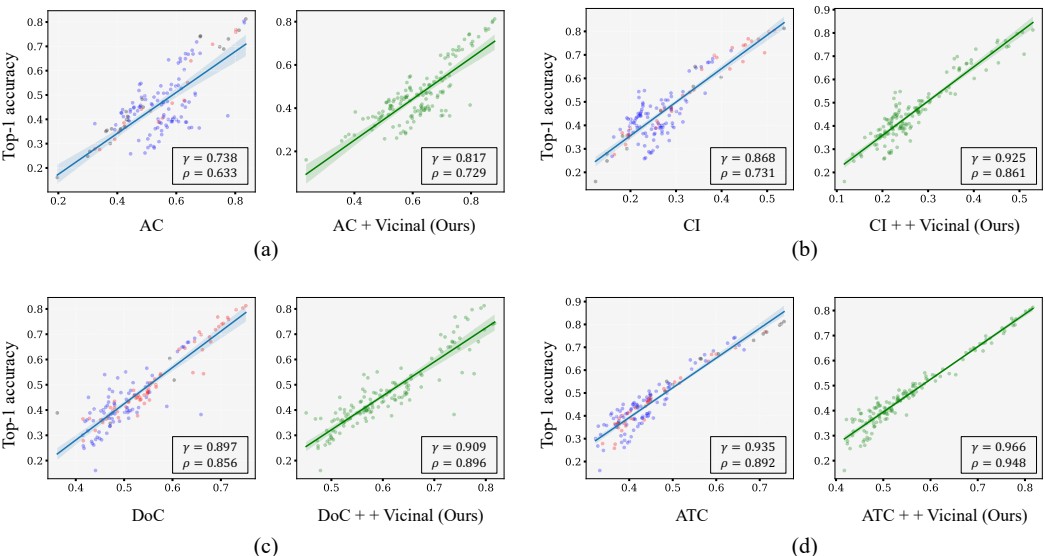

Figure F4: **Correlation between various risk proxies and accuracy on the ImgeNet-R dataset.** All notations in each figure have the same meanings as Fig. 5 of the main ppaer. We observe that proposed vicinal assessment effectively rectifies the risk estimates for the majority of models under various risk proxies.

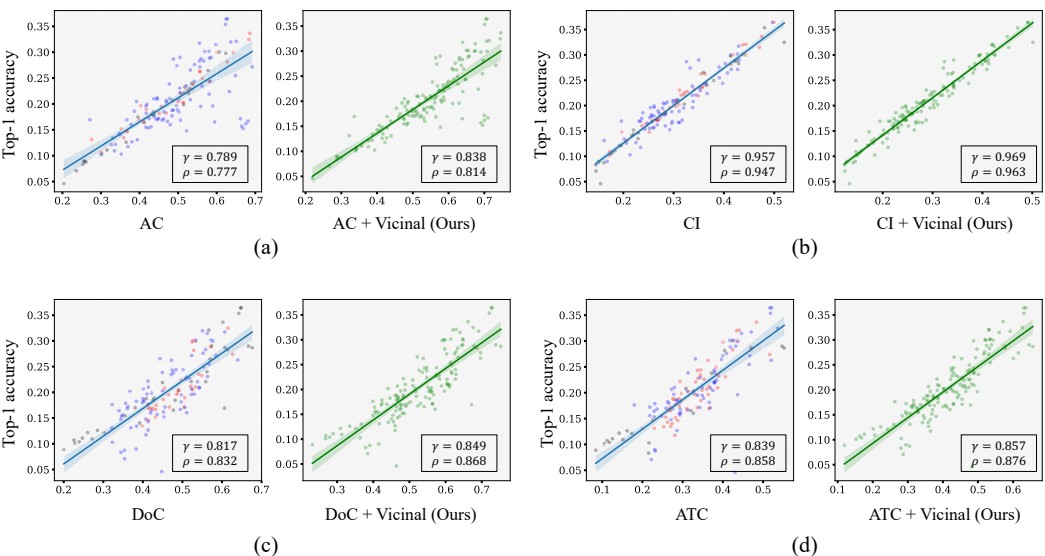

Figure F5: **Correlation between various risk proxies and accuracy on the ObjectNet dataset.** Notations in each figure have the same meanings as Fig. 5 of the main paper. Our observations are similar with those in Fig. F4.

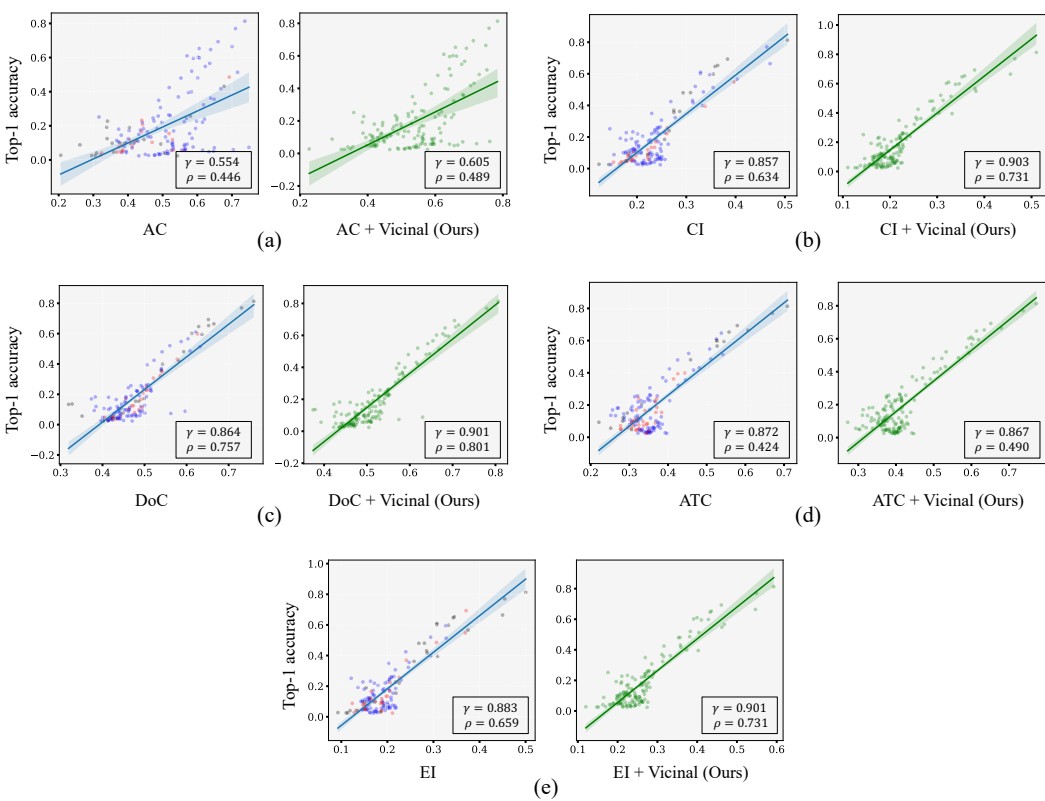

Figure F6: **Correlation between various risk proxies and accuracy on the ImageNet-A dataset.** Notations in each figure have the same meanings as Fig. 5 of the main paper. Our observations are similar with those in Fig. F4 and Fig. F5.