# OpenReview forum: "Vicinal Assessment of Model Generalization"
_ICLR.cc/2024/Conference — ICLR 2024 Conference Withdrawn Submission_

### Official Review · Reviewer_R4Lp · 2023-10-26

**Soundness:** 2 fair
**Presentation:** 3 good
**Contribution:** 2 fair
**Rating:** 5
**Confidence:** 4

**Summary:**

This paper explores the relationship between existing indicators equipped by vicinal assessment and the generalization ability of classification models on out-of-distribution test sets. It finds that re-weighting existing indicators by considering neighboring test samples can solve the imprecise generalization prediction caused by spurious model responses. The authors empirically show that their proposed universal method separates the softmax probability of correct and incorrect predictions farther than original methods, and theoretically certify that this property leads to a successful rank of classification models for the underlying generalization performance.

**Strengths:**

1. **[This paper is clearly written and well organized.]** Parts of the motivation, the approach introduction, and the experiments are easy to follow. Readers can clearly understand their goal, what they are doing, and why they are doing so.
2. **[The idea of this approach is reasonable.]** It is definite for current logit-based methods to encounter overfitting problems. Predicting softmax probability with considering neighbors is a reasonable method to alleviate this problem.
3. **[The comparison between model-centric and data-centric approaches in experiments is insteresting.]** My favorite part of this paper is the discussion about the prediction of generalization performance on different setups. In practice, the two setups have their own application scenarios, giving rise to the importance of exploring them both.

**Weaknesses:**

1. **[The proposed method has limited application scenarios.]** The proposed method is designed to equip logit-based methods only, which means other types of methods, such as ProjNorm [1], Dispersion Score [2], and COT [3], cannot apply it.
2. **[Some discussions are not explored deeply.]** The first is the discussion about the model-centric and data-centric approaches. In detail, in the third part of the discussion section and the relevant supplementary material, you do not provide reasons why your proposed method cannot work well under the data-centric setup. The second is that you only discuss the enhancement connection between your proposed method and logit-based methods, but ignore the parallel relationship between your proposed method and non-logit-based methods. For example, both Dispersion Score and COT mention that they can avoid overfitting or miscalibration problems, so what is the advantage of your method?
3. **[Theoretical insights are unconvincing.]** In my view, the key point of this theoretical part is to certify why your proposed method can separate correct and incorrect samples in the softmax-probability space, rather than to certify that your method can successfully rank classification models based on their generalization performance with the separability property.

[1] Yu, Yaodong, et al. "Predicting out-of-distribution error with the projection norm." International Conference on Machine Learning. PMLR, 2022.

[2] Xie, Renchunzi, et al. "On the Importance of Feature Separability in Predicting Out-Of-Distribution Error." arXiv preprint arXiv:2303.15488 (2023).

[3] Lu, Yuzhe, et al. "Characterizing Out-of-Distribution Error via Optimal Transport." arXiv preprint arXiv:2305.15640 (2023).

**Questions:**

1. Can you also discuss the relation to other types of approaches, such as ProjNorm, Dispersion Score or COT?
2. Can you further explain why your approach cannot work well under the data-centric setup?
3. Can you explain why your approach can separate correct and incorrect samples apart?
4. In the working mechanism in Page 4 of supplementary materials, can you explain why $\frac{1}{m}\sum_{i=1}^m p_v^i>\frac{1}{m}\sum_{i=1}^m p_e^i \Rightarrow P(\frac{1}{m}\sum_{i=1}^m \widehat{R_v}(f_a,x_i)-\frac{1}{m}\sum_{i=1}^m\widehat{R_v}(f_b,x_i)>0)> $

$P(\frac{1}{m}\sum_{i=1}^m \widehat{R_e}(f_a,x_i)-\frac{1}{m}\sum_{i=1}^m\widehat{R_e}(f_b,x_i)>0)$?
Why $\frac{1}{m}\sum_{i=1}^m \widehat{R_e}(f_a,x_i)=\widehat{R_e(f_a)}$ with the condition of $m\neq n$?

---

### Official Review · Reviewer_fA71 · 2023-10-28

**Soundness:** 3 good
**Presentation:** 4 excellent
**Contribution:** 1 poor
**Rating:** 3
**Confidence:** 5

**Summary:**

This paper proposes a method called vicinal risk proxy (VRP) to estimate the confidence (vicinal score) of a sample by testing the samples around it. However, the innovation of this work is insufficient, as it uses a very straightforward method and similar ideas have appeared in many previous works (including calibration). The author's explanation of the work is very detailed, and the formulas are very clear, but the method does not appeal to me.

**Strengths:**

- The author provides very clear examples to introduce their method, which allows me to quickly understand the work done in this paper. In particular, Figure 1 helps me to understand the work done in this paper and how it is implemented as soon as I see it.

- The author's formulas are very clear, especially the repeated declarations of symbols in each formula, which prevents me from having to flip back to the beginning of the paper to find the definition of a symbol when I encounter it later in the paper.

**Weaknesses:**

The innovation of the paper is insufficient. In fact, as early as the beginning of 2020, I had tried a similar idea, which is almost identical to the approach described in this paper, and the paper only describes my first version of the attempt. Here is my experience and why I did not continue this research:


- Calculating dµ involves a significant amount of additional time, so I abandoned this approach since the performance of ensemble methods can surpass many calibration methods, and their biggest drawback is the substantial inference cost. If a method cannot outperform an ensemble in terms of time efficiency, its performance needs to surpass that of the ensemble, but I found that the performance of this method cannot exceed the ensemble method.

- To address this issue, I used a neural network to fit the similarity of samples, i.e., training an additional branch on the original network to fit the vicinal score, which transfers the additional time cost to the training process and significantly reduces the time spent during the inference process. However, this method is still unsatisfactory for two reasons:

  - 1. Confidence suffers from severe overfitting, as the confidence of the vast majority of samples in the training set is 1.

  - 2. The proxy estimation of confidence depends on the surrounding samples, which means that it requires good confidence estimation of the surrounding samples, so this estimation does not fundamentally solve the problem.

- Finally, I changed my strategy and used true class probability (TCP[1]) as the confidence of the vicinal sample, which can effectively alleviate the problem of poor confidence estimation of the vicinal sample, to some extent solving the issues.

However, I ultimately did not continue to study this work because the method is not elegant enough.

_[1] Corbière C, Thome N, Bar-Hen A, et al. Addressing failure prediction by learning model confidence[J]. Advances in Neural Information Processing Systems, 2019, 32._

**Questions:**

I have no other questions about this paper. I hope the author can continue to conduct more in-depth research, and the previous research I just described can provide useful information for the author.

---

### Official Review · Reviewer_FoiT · 2023-10-31

**Soundness:** 3 good
**Presentation:** 4 excellent
**Contribution:** 3 good
**Rating:** 6
**Confidence:** 4

**Summary:**

The paper considers the problem of evaluating model generalization on out-of-distribution datasets without having access to the ground truth labels (e.g., if we wish to rank a set of models based on their eventual model accuracy on the OOD test set).

A key observation is that prior works which compute the score based on individual samples suffer in cases when a prediction made with high confidence by the model is wrong (or vice versa, when low confidence prediction is correct) which brings noise in the evaluation. Borrowing ideas from the general concept of vicinal risk minimization, the authors propose vicinal assessment by integrating model responses of neighboring samples from the test set when computing the score for a given test sample $x$.

The authors show that their methodology can be applied on top of prior works demonstrating its benefits on a wide range of existing models and a few datasets.

**Strengths:**

* Well written paper. Relatively easy to read and follow. Evaluation seems thorough and rigorous.
* Builds on top of (and can be integrated with) prior works, improving their results.
* Creatively proposes the usage of vicinal risk in the context of unsupervised evaluation.

**Weaknesses:**

I believe that most of the weaknesses are addressed, mostly in the appendix, but the relevant sections are not properly referenced from the main paper.

* Computational cost and runtime (App. D)\
In Table D1 it is not clear if running time (seconds) is total running time for the whole test set, or average per sample. Somehow, the numbers don't add up. It is not clear what is the value of $m$ that you use. If you set, e.g., `m = 50`, one would expect 50x slowdown (per sample on average) unless you perform some caching (e.g., using the same transformed images all the time).
* I would appreciate if you could elaborate on how you select the neighboring examples (or at least reference App. D where you describe this). This would also prevent confusion with footnote 1 why "Because of the limited size of ImageNet-R, 150 is the maximum value we can set $m$ to". And also how do you select the neighboring samples when there are more than $m$ with the same predicted label?

**Questions:**

Q1: Does it make sense to also include the scores of the evaluated samples themselves (e.g., $x_1$ and $x_2$ in Fig. 1)? Does it change anything quantitatively or doesn't really matter?

Q2: "Vicinal assessment is neither beneficial nor detrimental on near-OOD test sets" - could you, please, provide any insights why this is the case?

Q3: Can you combine / learn different similarity functions and transformations? If the impact of the different image transformations is limited (as we observe in Table 2), can we (and should we) select them arbitrarily? Is there a reason some transformations might be more beneficial than others and in what settings would this be the case? How should these design decisions be made in a general setting?

Q4: Do you apply the same transformation to a give sample all the time, or sample it randomly each time? What happens if you apply and aggregate multiple transformations simultaneously (e.g., grey-scale + color jitters + rotation)?

Minor: Should it be $\ell(f(\mathbf{x}), y)$ in Eq. (5) ? I.e., drop the index $i$ from $y_i$?

---

### Official Review · Reviewer_TbRp · 2023-11-01

**Soundness:** 3 good
**Presentation:** 2 fair
**Contribution:** 2 fair
**Rating:** 3
**Confidence:** 3

**Summary:**

The generalization ability on out-of-distribution test sets has been assessed as confidence or invariance, which are calculated in isolation. Since it can be subject to spurious model response, this paper suggests utilizing the output of neighboring test samples. In other words, this paper suggests a new way of measuring model generalization ability on ood data.

**Strengths:**

- This paper suggests a new method to measure the generalization ability of a model to ood data which is less affected by the spuriousness or overconfidence.

**Weaknesses:**

- Since it requires knowing the output of neighboring samples and the similarity scores between samples, it may require more time and calculation. How much?
- I got the idea that using the confidence or invariance of a single test sample naively can be wrong. I wonder why using vicinal risk solves the problem.

**Questions:**

- Why it is limited if it is applided to IND? I saw that models generally have good behaviours, but considering the calibration or overconfidence issue, IND is also not free from those. Then what makes the difference between IND and OOD?
- How can we approve that the outputs of the neighboring samples are correct? If it cannot be proved, what is the benefit of using neighbor samples?
- How the single test sample were selected for figure 2? Is this pattern shown for any arbitrary test sample?
- Supplementary material is supposed to be following the main paper, as the conference guide mentioned.